# Transcriptomic Insights: Phytogenic Modulation of Buffel Grass (*Cenchrus ciliaris*) Seedling Emergence

**DOI:** 10.3390/plants13091174

**Published:** 2024-04-23

**Authors:** Xipeng Ren, Tieneke Trotter, Nanjappa Ashwath, Dragana Stanley, Yadav S. Bajagai, Philip B. Brewer

**Affiliations:** Institute for Future Farming Systems, Central Queensland University, Rockhampton, QLD 4701, Australia; xipeng.ren@cqumail.com (X.R.); t.trotter@cqu.edu.au (T.T.); y.sharmabajagai@cqu.edu.au (Y.S.B.)

**Keywords:** phytogen, RNA-seq, seedling emergence, grass, pasture, transcriptome, metabolism

## Abstract

This study explores the impact of a novel phytogenic product containing citric acid, carvacrol, and cinnamaldehyde on buffel grass (*Cenchrus ciliaris*) seedling emergence. A dilution series of the phytogenic solution revealed a concentration range that promoted seedling emergence, with an optimal concentration of 0.5%. Transcriptomic analysis using RNA-seq was performed to investigate gene expression changes in seedlings under the influence of the phytogenic product. The results revealed that the phytogenic treatment significantly altered the gene expression, with a prevalent boost in transcriptional activity compared to the control. Functional analysis indicated the positive alteration of key metabolic pathways, including the tricarboxylic acid (TCA) cycle, glycolysis, and pentose phosphate pathways. Moreover, pathways related to amino acids, nucleotide biosynthesis, heme biosynthesis, and formyltetrahydrofolate biosynthesis showed substantial modulation. The study provides valuable insights into the molecular mechanisms underlying the phytogenic product’s effects on grass seedling establishment and highlights its ability to promote energy metabolism and essential biosynthetic pathways for plant growth.

## 1. Introduction

Phytogens or phytogenic products is a term that describes the natural compounds produced by plants [1]. Phytogenics comprise a wide range of substances, such as hormones, pigments, antioxidants, essential oils, etc. [2]. Aside from their common usage in livestock feed to improve animal performance via antibiotic-free antimicrobial effects [3], these plant-derived products have various benefits on plant growth, development, and plant interactions with other organisms. One important aspect of plant phytogens is their potential to influence the early growth of seedlings.

Seedling emergence and early growth is a complex and crucial developmental transition in a plant’s life cycle [4]. This process is influenced by various internal and external factors, such as moisture, light, oxygen, and temperature [5]. Plant hormones and phytogens also modulate various physiological and biochemical processes to enhance or inhibit the emergence process, which depends on their type and concentration. Some plant compounds, such as abscisic acid (ABA) and phenolic acids, can induce seed dormancy and prevent emergence by inhibiting the activity of enzymes and hormones that promote emergence [6]. On the other hand, others, such as gibberellin (GA) and cytokinin, induce the synthesis of hydrolytic enzymes that facilitate the breakage of endosperm and the seed coat, and mobilize seed storage reserved in the endosperm that supports embryo growth, eventually promoting seedling growth [6]. Then, the subsequent growth and emergence of the seedling from soil involves changes in metabolism to support rapid growth along with the mechanical properties required to push through soil [7,8,9].

RNA-seq is a high-throughput sequencing technique that uses next-generation sequencing to identify the presence and quantity of RNA molecules, also known as transcriptome sequencing [10]. It can indicate the changes in gene expression during different developmental stages or treatments. Subsequent changes in metabolic and energy production pathways have been observed during the transition to rapid growth, which have been positively associated with the gene and gene expression changes that underlie seedling vigor [8,11]. RNA-seq can identify the differentially expressed (DE) genes between different treatments. It also reveals the pathways and networks involved in seedling growth and development, including plant hormone signaling, stress response, metabolism, and cell cycle. Therefore, RNA-seq is a valuable tool for understanding the mechanisms behind grass seedling vigor and the possible actions of phytogens.

In our previous research, we applied a phytogen-based product (PHY) to severely damaged pastures and investigated the effect of this product on pasture yield [12]. In addition to improvements in plant morphology and biomass, the application of PHY improved the soil microbiome by enriching the pathways involved in soil detoxification and carbon sequestration [12,13]. This commercial product contains citric acid, carvacrol, and cinnamaldehyde, which are natural compounds that belong to different classes of secondary metabolites, such as organic acids, phenols, and aldehydes [14]. Citric acid, carvacrol, and cinnamaldehyde have numerous biological activities, such as antioxidant, antimicrobial, antifungal, and herbicidal properties. Citric acid can affect plant growth and seeding emergence by influencing the pH, nutrient availability, and microbial activity of the soil. Citric acid can also act as a chelating agent, binding to metal ions and enhancing their uptake by plants. Therefore, it can help with stress tolerance, such as acidity, drought, salinity, and heavy metals [15].

Carvacrol is a phenolic compound found in several essential oils of aromatic plants [16]. Carvacrol can affect plant growth and seedling emergence by modulating the expression of genes related to stress response, photosynthesis, and cell division. Carvacrol presents antimicrobial properties that make it helpful for controlling pathogenic fungi and bacteria [17,18]. Likewise, cinnamaldehyde interferes with the synthesis of ergosterol, a vital component of fungal cell membranes [19].

To investigate if the plant-biomass-promoting effects of the phytogen could be derived via mechanisms involving the improvements in the rate of seedling establishment, we performed a set of experiments to determine if such effects exist and to find a range of optimal PHY concentrations. Although investigating the benefits of citric acid, carvacrol, and cinnamaldehyde individually would contribute to this study, the existing vast knowledge of their individual benefits did not result in a large-scale agricultural application perhaps due to their volatility or instability. Moreover, the mixes of different bioactive molecules can behave differently from the individual ingredients.

PHY used in the present study was initially developed as an antibiotic alternative for livestock delivered via drinking water. For that purpose, it was designed to overcome volatility, fast evaporation, and stability issues via a gentle yet robust emulsion delivery system, and it is already widely distributed globally, providing a range of established benefits to animal health [3]. A previous study has demonstrated that it has the capability to persist in improving soil for up to 18 months [12]. Thus, repurposing this plant-based product for use in pastures would fit current global goals of sustainability and reprocessing, with a product beneficial for nature, soil, animal, and plant life.

After confirming the ability of the phytogen to significantly increase the rate of seedling emergence, we proceeded into a transcriptomic study aiming to identify the influence of the phytogen-based product on gene expression during the seed emergence process.

## 2. Results

### 2.1. Emergence Record and Selection of Optimal Concentration

The emergence rate under a dilution series across four weeks is shown in Figure 1. The high concentrations of PHY, including 2%, 5%, and 10%, entirely prevented seedling emergence.

Figure 1 depicts the emergence rate of buffel grass in sand over 28 days, with different lines representing various concentrations of PHY. The control (CTR) line showed a steady increase in emergence rate over time, reaching 16.67% at day 28. The 0.05% of PHY had a similar trend but at a higher rate, which was 25% in the end. Both 0.1% of PHY and 0.5% of PHY lines showed an initial increase, then plateau around day 18, with the latter having a higher emergence rate overall and finally reaching 30%. The 1% of PHY had the lowest emergence rate (11.67%), showing little growth over time. Higher concentrations (2%, 5%, and 10%) showed inhibitory effects and prevented emergence. In this longitudinal study, 0.5% of PHY significantly changed emergence over the time period (*p* = 0018, Wilcoxon matched-pairs signed rank *t*-test), with a significant positive effect from day 17 (*p* = 0.0249, Unpaired *t*-test). 

All the concentrations shown in Figure 1, except for 1%, positively impacted emergence rates; however, they varied in steepness, perhaps indicating different growth rates. The control and 0.05% of PHY groups exhibited consistent growth throughout the period, while 0.1% of PHY and 0.5% of PHY plateaued after approximately day 18. Notably, compared with other dilution series, 0.5% of PHY revealed the highest emergence rate from day 12, and then it kept the trend until the end of the experiment, suggesting that the buffel grass had the best emergence performance under this concentration. Therefore, 0.5% of the phytogenic product was selected for further analysis.

### 2.2. Transcriptomic Sequencing Quality Control

We performed RNA-seq analysis using *Oryza sativa* (35,806 genes) as a reference genome. RNA was extracted from whole seedlings of 2 cm in shoot height. The control group consisted of sample G1, G2, G3, G4, and G5, while the treatment group included G6, G7, G8, G9, and G10.

Overall, the number of mapped sequences was 55–60% in all samples, providing us with information on buffel grass transcripts used in the present analysis. After filtering out the unmapped reads and mapped reads in broken pairs which were not properly aligned to the reference genome, the treatment yielded 178.4 million bp sequences mapped against the reference genome total, with an average of 35.7 million bp per sample, accounting for 43.2% of raw paired reads. Similarly, the control group generated a total of 175.9 million bp sequences, with an average of 35.2 million bp per sample, constituting 46.6% of raw paired reads. The Mann–Whitney test showed that there was no significant difference in the depth of sequencing in two groups (*p* = 0.15).

### 2.3. Differential Expression in Seedling Emergence

Upon the gene expression (GE) tracks from RNA-seq, a PCA plot was built as shown below (Figure 2), describing the sample distribution across the first two components, with a 95% confidence interval, and each line connects each point to the centroid. It reveals a distinct separation between control and phytogenic treatment, highlighting the significant effect that PHY had on the samples. Additionally, within the groups, PHY had less variability among all the samples compared to the control, implying that PHY had a homogenizing effect on the samples, leading to a more consistent response among individuals.

Among all the 35,806 genes from mapped sequences, 5261 differentially expressed genes passed the filter with an absolute fold change greater than 1.5 and a statistical significance threshold of *p* < 0.05. The heatmap (Figure 3) further illustrates the contrasts between the two groups, where the genes with an increased expression are separated into two clusters, colored with a red hue, underlining the differential gene expression profiles between control and treatment across the top 100 DE genes.

The differential gene expression is also visualized via a volcano plot (Figure 4). Upregulated genes (fold change > 1.5, *p* < 0.05) from PHY are presented by red dots, reaching a log_2_ fold change greater than 0.585, while the blue dots signify downregulated genes (fold change < −1.5, *p* < 0.05), which had a log_2_ fold change less than −0.585. The rest of the genes under this filtering criteria are colored with gray. 

Having established the sequencing data quality, we performed a functional analysis to obtain the gene expression difference between control and phytogenic treatment.

### 2.4. Pathway Analysis in Seedling Emergence

According to the fold change and *p*-value cutoff, 5261 DE genes from PHY were divided into two groups, including 2938 upregulated genes with a fold change > 1.5, *p* < 0.05 and 2323 downregulated genes with a fold change < −1.5, *p* < 0.05. All the DE genes were uploaded into the PANTHER Classification System, in which 2856 increased genes and 2268 decreased genes were annotated with a reference genome. By uploading annotated DE genes into the functional classification analysis, we generated the pathways and their associated DE genes identified in PHY. More than half of the pathways could be predicted to be activated by the phytogen where PHY promoted more genes to highly express compared to the control (*p* < 0.05) (Appendix A), including 5-hydroxytryptamine biosynthesis, chorismate biosynthesis, cytoskeletal regulation by Rho GTPase, de novo purine biosynthesis, formyltetrahydrofolate biosynthesis, heme biosynthesis, isoleucine biosynthesis, leucine biosynthesis, lysine biosynthesis, pyruvate metabolism, salvage pyrimidine ribonucleotides, and TCA cycle. However, other pathways have more complex regulatory mechanisms (Appendix A), including 5-hydroxytryptamine degradation, apoptosis signaling pathway, cadherin signaling pathway, de novo pyrimidine ribonucleotides biosynthesis, folate biosynthesis, glycolysis, pentose phosphate pathway, threonine biosynthesis, and tryptophan biosynthesis. In these pathways, some genes were significantly activated by PHY, while a considerable number of genes were also remarkably inhibited (*p* < 0.05).

Of all the pathways that have more genes activated by PHY, the Tricarboxylic Acid (TCA) cycle is related to the metabolism of carbohydrates which provides the energy required for the rapid growth process (Figure 5). It comprises a series of chemical reactions where four genes were significantly highly presented by PHY (*p* < 0.05). They encode four catalytic enzymes, pyruvate dehydrogenase, citrate synthase, succinate--CoA synthetase, and malate dehydrogenase, which individually modulate four different reactions in this pathway. Four reactions enriched in DE genes upstream and downstream of the TCA cycle imply that PHY might have an influence on those reactions.

Another carbohydrate-relevant pathway is pyruvate metabolism which shares some components with the TCA cycle and provides pyruvate as the substrate. Therefore, in the phytogen treatment, pyruvate metabolism also had a few of the genes activated, encoding malate dehydrogenase, citrate synthase, ATP-citrate synthase beta chain protein 1, and pyruvate dehydrogenase. Notably, PHY also inhibited two genes, one of which regulates citrate synthase (Appendix A). 

In addition to the above-mentioned preference of the PHY for activation, it also exhibited inhibition in some carbohydrate metabolism. As the substrate of the TCA cycle, pyruvate is also produced from glycolysis where PHY displayed both activation and inhibition (Figure 6). There were two genes significantly increased in PHY and two genes were inhibited (Appendix A) (*p* < 0.05). Both inhibitory genes regulate glucose-6-phosphate isomerase upstream of the pathway, while triosephosphate isomerase was activated, and, in downstream reaction, the highly expressed gene *OsPK2* regulates the activity of pyruvate kinase to produce pyruvate as the last step of glycolysis.

PHY also reduced more genes in the pentose phosphate pathway (Appendix A). There were three genes significantly reduced by PHY, two of which encode fructose-6-phosphate biosynthesis, while only one gene was increased, targeting transaldolase (Appendix A). Both catalytic reactions have a common intermediate product, implying that PHY might modulate fructose-6-phosphate biosynthesis in a balanced way.

Seedling emergence and growth also requires the synthesis of complex molecules, such as amino acids and nucleotides. In PHY treatment, isoleucine, leucine, and lysine biosynthesis all have genes significantly increased (Appendix A). Typically, the essential animo acid lysine biosynthesis involves five genes that were activated by PHY, although two other genes were inhibited. On the other hand, PHY prevented two genes participating in tryptophan biosynthesis, controlling the last two steps of the pathway, while only one gene was identified to be highly expressed in the PHY group (Appendix A).

PHY significantly increased a few genes involved in nucleotide metabolism. One of them is de novo purine biosynthesis which has seven genes that are responsible for seven enzymes increased by PHY, including UMP-CMP kinase 4, adenyl succinate synthetase 2, guanylate kinase 2, adenylate kinase 3, nucleoside diphosphate kinase, formyltetrahydrofolate deformylase family protein, and adenylate monophosphate kinase 5. Another two genes encoding the same enzyme, phosphoribosyl glycinamide formyl transferase 1, were decreased by PHY (Appendix A). Similarly, PHY also promoted three genes in salvage pyrimidine ribonucleotides but no gene was decreased in this pathway. On the other hand, some nucleotide-related metabolism was also affected by the inhibition of PHY. De novo pyrimidine ribonucleotide biosynthesis has three DE genes, one of which was negatively influenced by PHY. Folate biosynthesis has been identified to have two genes increased by PHY and two genes were decreased (Appendix A).

Furthermore, PHY also significantly altered genes in other pathways. One of them is heme biosynthesis in which seven genes were overrepresented and one gene was less expressed when adding PHY (Figure 7). Five catalytic enzymes, including glutamyl-tRNA-synthetase, glutamate-1-semialdehydeaminot, porphobilinogen synthase, hydroxymethylbilane synthase, and uroporphyrinogen decarboxylase, are encoded by seven increased genes (*p* < 0.05). Only uroporphyrinogen methyltransferase was inhibited. The significant high expression of these upstream reactions might also have an influence on the downstream reactions of heme biosynthesis. 

In addition, both chorismate biosynthesis and formyltetrahydrofolate biosynthesis have genes that were promoted by PHY, with no gene inhibited (Appendix A). In chorismate biosynthesis, three genes were shown to be highly expressed due to the application of PHY, targeting two enzymes, 3-dehydroquinate synthase and shikimate kinase (Appendix A). In formyltetrahydrofolate biosynthesis (Appendix A), two genes involved in dihydrofolate synthase and formyltetrahydrofolate deformylase were activated by PHY.

PHY may also influence hormone activation. In the *LOG* (*LONLEY GUY*) gene family, *LOG1*, *LOGL4,* and *LOGL7* were significantly upregulated in PHY treatment (Appendix A). This gene clade makes bioactive cytokinin. They encode cytokinin riboside 5′-monophosphate phosphoribohydrolases that convert inactive cytokinin to the biologically active free-base forms.

Generally, PHY showed more activation than inhibition in most metabolism pathways, such as carbohydrates, amino acids, and nucleotides, during seedling growth and development. It also exhibited upregulation in genes related to cytokinin activation. However, PHY also displayed a complex regulation in some pathways where a considerable amount of promoted and inhibited genes was observed.

## 3. Discussion

Seedling emergence is a crucial stage in plant development and can be considered as a determinant for plant productivity. Seeds are activated from dormancy, utilizing the stored nutrients to support the growth of the embryo and the establishment of the seedling [20]. This experiment was to explore the alteration of gene expression during emergence, providing insights into the potential benefits of phytogenic products. The concentration of the phytogenic solution would have various effects on seedling growth. A low concentration of a phytogen can stimulate seedling emergence; however, the high concentration of a phytogen can inhibit it [21]. A phytogen may interact with plant hormones, including ABA and GA, to regulate seedling emergence, in which ABA is known to slow down the process but GA promotes it [22]. As is shown in this study, 2%, 5%, and 10% of PHY treatment displayed a completely inhibitory effect. Although some seeds emerged when applying 1% of PHY, they had the lowest establishment rate in the end. Among all the lower concentrations, 0.5% was the highest but stimulated the seed to have the highest emergence rate, suggesting that this process needs an appropriate phytogen concentration, and too low of a concentration is not conducive to it.

Through a detailed analysis of the transcriptomic profile in the seedling samples, the benefits of the phytogenic product were displayed at the transcript level. The RNA-seq analysis revealed that half of the DE genes were activated, and the other half were inhibited by PHY, suggesting the big difference that PHY made to the gene expression during seeding growth and development. Based on the separation of genes, our pathways analysis further delved into the significant effect that PHY had on metabolism. 

Firstly, seedling emergence needs a series of reactions to release energy from the metabolism of organic matter. The TCA cycle, also known as the citric acid cycle, that is related to carbohydrate metabolism, was identified to have four DE genes highly expressed in the PHY group (Figure 5). This energy metabolism occurs in the mitochondria of eukaryotic cells, as a central metabolic pathway for all the organisms, and plays a crucial role in the breakdown of organic matters, such as glucose, fatty acid, and amino acid to produce energy [23]. During emergence, it also provides carbon skeletons to anabolic processes and contributes to carbon–nitrogen interaction and biotic stress responses [24]. The TCA cycle is catalyzed by different enzymes. Pyruvate dehydrogenase converts pyruvate, from glycolysis, into acetyl-CoA, which then enters the TCA cycle [25]. During emergence and seedling growth, pyruvate dehydrogenase ensures the efficient conversion of pyruvate to acetyl-CoA, fueling the TCA cycle and supporting growth and development. Citrate synthase catalyzes the first step of the TCA cycle where it combines acetyl-CoA with oxaloacetate to form citrate [26]. As the initiator of the TCA cycle, it ensures a continuous supply of citrate, which serves as a precursor for various biosynthetic pathways. Succinyl-CoA Synthetase catalyzes the conversion of succinyl-CoA to succinate, with the production of GTP that can be converted to ATP to support various cellular processes during emergence and seedling growth [27]. Malate dehydrogenase, encoded by the DE gene Os08g0562100, also plays a critical role in the TCA cycle in emergence. It catalyzes the reversible conversion of malate to oxaloacetate using the reduction of NAD^+^ to NADH + H^+^ (Figure 1). Then, oxaloacetate is converted to citrate, catalyzed by citrate synthase. The cycle then proceeds through a series of reactions to produce NADH, FADH2, and ATP [28].

Another pathway related to energy release is glycolysis. Differing from the TCA cycle, PHY did not significantly increase as many genes as in the TCA cycle; instead, it affected this metabolism in both activation and inhibitory ways (Figure 6). Glucose-6-phosphate isomerase, responsible for the interconversion of glucose-6-phosphate and fructose-6-phosphate [29], has two genes inhibited by PHY; however, triosephosphate isomerase was activated and pyruvate kinase was also promoted, which catalyzes the final step of glycolysis, the conversion of phosphoenolpyruvate to pyruvate, with the concomitant production of ATP [30]. As the initial substrate of the TCA cycle, pyruvate underscores the importance of glycolysis for seedling emergence [31]. However, unlike the TCA cycle, it is hard to predict whether PHY had a positive or negative impact on glycolysis due to its mix of regulations, but we could know that enzymes having increased genes would benefit other metabolisms.

Similar to glycolysis, the pentose phosphate pathway also has genes that were both increased and decreased by PHY (Appendix A). The pentose phosphate pathway occurs in the cytosol of plant cells and is involved in the production of NADPH and pentoses, which are used in the biosynthesis of nucleotides, amino acids, and other important biomolecules [32]. Transaldolase activated by PHY plays a crucial role in the pentose phosphate pathway. It catalyzes the transfer of a three-carbon unit from sedoheptulose-7-phosphate to glyceraldehyde-3-phosphate, producing erythrose-4-phosphate and fructose-6-phosphate [33]. Therefore, transaldolase links the pentose phosphate pathway to glycolysis as fructose-6-phosphate produced by transaldolase can be converted to glucose-6-phosphate, and then enter glycolysis. The other product, erythrose-4-phosphate, could serve as a precursor in the biosynthesis of the aromatic amino acids, such as tyrosine, phenylalanine, and tryptophan [34]. However, fructose-6-phosphate production is also regulated by glucose-6-phosphate isomerase, in which two encoding genes were significantly reduced by PHY. Therefore, fructose-6-phosphate would be accumulated due to the activation of transaldolase, promoting more fructose-6-phosphate synthesis, while the inhibitory effect of glucose-6-phosphate isomerase prevented the conversion of fructose-6-phosphate and glucose-6-phosphate. The abundance of fructose-6-phosphate could benefit glycolysis to provide energy for seedlings.

The energy produced would be used in various cellular components and biological processes during seedling growth. Amino acids are the building blocks of proteins and other essential molecules in plants; especially during emergence, seeds need to synthesize new proteins for growth and development [20]. Among all the amino acids related to the genes promoted by PHY, lysine is a precursor for the biosynthesis of other important compounds such as alkaloids, hormones, and secondary metabolites. Lysine is also involved in the regulation of plant–microbe interactions, such as the nodulation of legumes by rhizobia [35].

Nucleotide biosynthesis is another crucial biological process during emergence and seedling growth because cell division, as well as growth, needs DNA replication and transcription. Moreover, nucleotides participate in energy transfer and signaling pathways [20]. In this study, PHY significantly increased seven genes involved in the de novo purine biosynthesis pathway (Appendix A). Of all the enzymes encoded by seven genes, UMP-CMP kinase 4 catalyzes the phosphorylation of UMP and CMP to form UDP and CDP, respectively [36]. Nucleoside diphosphate kinase catalyzes the transfer of a phosphate group from a nucleoside triphosphate (NTP) to a nucleoside diphosphate (NDP), producing a new NTP and NDP [37]. The formyltetrahydrofolate deformylase family protein catalyzes the hydrolysis of 10-formyltetrahydrofolate (formyl-FH4) to FH4 and formate [38]. Adenylate kinase 5 catalyzes the reversible transfer of phosphate groups between ATP and AMP, producing two molecules of ADP [39]. Adenyl succinate synthetase 2, encoded by gene *PURA2*, catalyzes the GTP-dependent conversion of inosine monophosphate (IMP) and aspartic acid to guanosine diphosphate (GDP), phosphate, and N-6-1,2-dicarboxyethyl-AMP [40]. The joint action of enzymes regulated by highly expressed genes ensured the synthesis of purine nucleotides, which are essential for DNA and RNA synthesis, energy metabolism, and signaling pathways in emergence [41].

In addition, resisting external environmental pressure also requires energy. Heme is an iron-containing porphyrin compound and recent study has shown that heme participates in the exogenous aminolevulinic acid (ALA)-promoted growth and antioxidant defense system of cucumber seedlings under salt stress [42]. The heme biosynthesis involved five enzymes encoded by seven activated genes in PHY (Figure 7), in which glutamate–tRNA ligase catalyzes the activation of glutamate to glutamyl-tRNA, and then glutamyl-tRNA is converted to 5-ALA by glutamate-1-semialdehydeaminot. Porphobilinogen synthase converts 5-ALA into porphobilinogen (PBG) and PBG is converted to hydroxymethylbilane. Uroporphyrinogen decarboxylase catalyzes and converts uroporphyrinogen III to coproporphyrinogen III [43]. Although these catalytic reactions occur upstream, we could predict that the downstream of heme synthesis might also be positively affected because one inhibitory gene was found in uroporphyrinogen methyltransferase, which means that more uroporphyrinogen III could be used for the conversion of coproporphyrinogen III (Figure 7).

Formyltetrahydrofolate biosynthesis is another essential pathway for plant growth and development. The formyltetrahydrofolate deformylase family protein controls the last step that converts 10-formyl-tetrahydrofolate (THF) to THF (Appendix A). THF is an essential cofactor in one-carbon metabolism, which is involved in the biosynthesis of nucleotides, amino acids, chlorophyll, and other important biomolecules [44]. It is also involved in nitrogen metabolism in plants [45]. In the PHY group, a formyltetrahydrofolate deformylase family protein gene was upregulated, which could promote the synthesis of THF and chlorophyll metabolism.

Cytokinin, as a crucial plant hormone synthesized mostly in roots and transported to the shoot, regulates the cell cycle and a range of developmental processes [46]. Two pathways have been involved in producing active cytokinin in plants: the two-step activation pathway and the direct activation pathway [47]. Our results show that the genes *LOG1*, *LOGL4,* and *LOGL7,* involving the direct activation pathway, were increased due to PHY treatment. The *LOG* gene family encodes cytokinin riboside 5′-monophosphate phosphoribohydrolases that release cytokinin nucleobase and ribose 5′-monophosphate. *LOG* hydrolyzes cytokinin riboside 5′-monophosphate but not AMP, suggesting that *LOG* is specifically involved in cytokinin activation [47]. PHY may promote plant growth by stimulating the production of active cytokinin in the root, which travels to the shoot and promotes cell proliferation in the meristem and leaves, potentially leading to accelerated seedling growth [48]. Hence, *LOG*-related genes could be an indicator of the increased growth promoted from roots. An increased *LOG*-related gene expression may be the way by which PHY promotes seedling emergence, and this would represent a target for future research.

Overall, under the activation of the phytogenic product, the seedlings not only may have accumulated energy through carbohydrate metabolism, but also efficiently used the energy to synthesize various substances required for growth and development. Notably, PHY had both activation and inhibition effects on the regulation of complex metabolic pathways, which suggests that, if we want to explore the expression effect of PHY on these pathways as a whole, we should not only consider the role of individual reactions, but also take account into each reaction comprehensively. However, this experiment focused on the role of genes under the significant influence of PHY in seedling emergence and seedling growth; therefore, we could only analyze the reactions and pathways involved in overrepresented genes, whether activated or inhibited, and then infer the expression effect of the whole metabolic process. Additionally, the number of DE genes is also affected by screening conditions. Simply, more DE genes could be selected under more relaxed screening conditions, which will make it easier to infer the role of PHY in the whole metabolic process.

## 4. Materials and Methods

### 4.1. Experiment Design

The commercial buffel grass (Gayndah) seeds were sourced from the *SELECTED SEEDS* company based in Pittsworth, QLD, Australia, and then stored at room temperature of 24 °C. Grass seeds were divided into two groups: control (CTR) and the phytogenic treatment Activo^®^ Liquid (EW Nutrition GmbH, Visbek, Germany) (PHY). The prepared coarse sand was washed and autoclaved.

To find the proper concentration of phytogenic product to promote seedling emergence, we designed a dilution series for each group, including 0% (control), 0.05%, 0.1%, 0.5%, 1%, 2%, 5%, and 10% from low to high. Each concentration group had three replicates with 20 seeds in each replicate. All the seeds were buried at a depth of 1 cm. For each replicate, approximately 500 g of autoclaved coarse sand was mixed with diluted phytogenic solution (20 mL), and then filled into the rectangular plastic container, taking up about 75% of the volume of the container. The control group received the same amount of water, setting the temperature at 30 °C during the day for 12 h and 25 °C at night for 12 h, using artificial light with a photosynthetic photon flux density (PPFD) of 100 µmol/m^2^/s. All the containers were placed in a growth cabinet indoors. Scoring for emergence started once all the seeds were sown and continued for 28 days. The best concentration was selected based on the emergence rate.

When the optimal concentration of the solution was selected, another batch of buffel grass seeds (20 seeds per group) was used to confirm the benefits of the selected concentration and obtain fresh seedling samples for RNA-seq analysis. The temperature and light condition would be the same as before.

The whole seedlings, including root and shoot, with a measured height of 2 cm were taken and stored at −80 °C for RNA-seq analysis. Both the control and treatment groups had five seedling samples (*n* = 5). Since the RNA content of each plant is difficult to extract, each of the five samples contained three plants.

### 4.2. Transcriptomic Sequencing

Five samples from each group were used to extract RNA using HiPure Plant RNA Mini Kit (Magen, Guangzhou, China). The quality was evaluated by NanoDropTMOneC (Thermo Fisher, Waltham, MA, USA). After extraction, 20 µL of RNA samples were transferred into RNA stabilization tubes (GENEWIZ from Azenta Life Sciences, Suzhou, China) and air-dried under the biosafety hood for 24 h. The stabilized RNA samples were shipped to Azenta Life Sciences (Suzhou, China) for sequencing.

Sequencing was carried out using a 2 × 150 bp paired-end (PE) configuration; image analysis and base calling were conducted by the Hiseq Control Software (HCS) + OLB + GAPipeline-1.6 (Illumina, San Diego, CA, USA) on the HiSeq instrument. A total of 10 buffel grass seedling samples (*n* = 5) from both control and phytogenic treatment were successfully extracted and sequenced. All sequence data are accessible on the NCBI Sequenced Read Archive (SRA) database with the accession number PRJNA1068799.

### 4.3. Data Analysis and Statistics

Calculating the emergence rate was carried out by dividing the number of seeds that sprouted by the total number of seeds tested and multiplying by 100. The result was exported into GraphPad Prism 10.1.0 [49] and plotted. The Wilcoxon paired *t*-test was applied to compare the statistic differences between the control and treatment [49].

Transcriptomic analysis was carried out using QIAGEN CLC Genomics Workbench 23.0.4 (QIAGEN, Aarhus, Denmark). Pair-end sequenced reads and metadata containing sample ID and treatment were imported to CLC Genomics Workbench, and recommended RNA-Seq Analysis workflow was used. The annotated *Oryza sativa* genome was downloaded from Public Repositories in CLC Workbench and selected as a reference. All annotated transcripts were extracted using mRNA track from the reference genome, and the reads were mapped against all the transcripts and to the whole genome. From this mapping, the reads were categorized and assigned to the transcripts using the Expectation Maximization (EM) algorithm and expression values for each gene were obtained by summing the transcript counts belonging to the gene. The transcripts per million (TPM) method was used to normalize the total counts. The quality control report (QC) would be generated as a part of the results. Data from the QC report was used in GraphPad Prism to perform Mann–Whitney statistic test to confirm no significant difference in sequencing depth between the two groups.

The Differential Expression (DE) function in CLC performed the statistical differential expression test using Gene Expression tracks (GE), which summarize expression at the gene level [50] generated from RNA-seq workflow. To visualize the differentially expressed genes, a heatmap with metadata was built upon *p*-value (*p* < 0.05) and fold change (absolute fold change of 1.5), presenting the top 100 DE genes. The principal component scatter plot (PCA) and volcano plot were inspected for the interpretation of RNA-seq.

The significantly altered genes were selected (*p* < 0.05 with an absolute fold change of 1.5) from DE result. The lists of significantly differentially expressed Gene IDs were imported to Protein ANalysis THrough Evolutionary Relationships (PANTHER) Classification System [51], and *Oryza sativa* was selected as the reference organism. The functional analysis was performed to find the relationship between DE genes and functional pathways.

## 5. Conclusions

Our study provides substantial insights into the role of this phytogenic product on gene expression during buffel grass seedling emergence, improving our understanding of its impact on seedling growth and development. The influence of PHY on seedling development was comprehensive, from improving emergence, affecting energy conversion pathways, to potentially synthesizing substances needed for development, such as cytokinin. Although the effects on most metabolic pathways were limited to specific reactions, it still gave us confidence that the potential benefit of this phytogenic product in seed emergence is worth applying in the future. Our study opens new avenues for deeper investigations into how PHY promotes seedling germination and vigor. Areas for further research have been revealed, including specific hormones, nutrients, metabolites, and enzymes that may be involved in seedling energy production and growth, along with how specific concentrations of PHY may affect a broader range of pasture and crop species, and how PHY may be employed to improve farming productivity.

## Figures and Tables

**Figure 1 plants-13-01174-f001:**
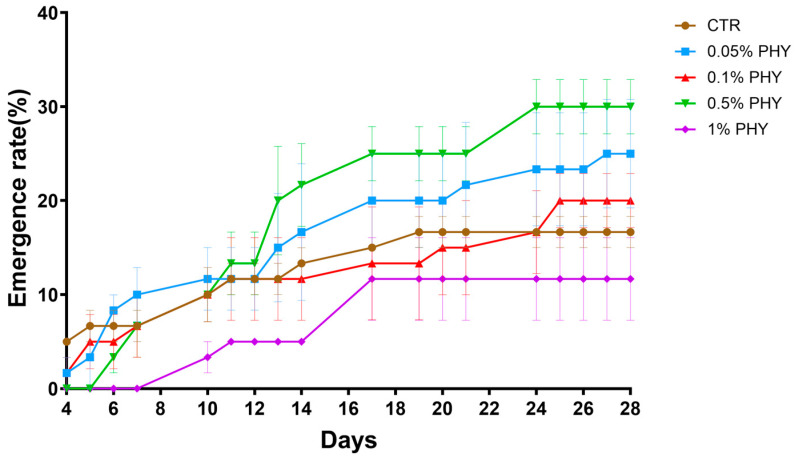
Changes of buffel grass seed emergence rate with time. Each symbol represents the mean emergence rate ± SEM, *n* = 3.

**Figure 2 plants-13-01174-f002:**
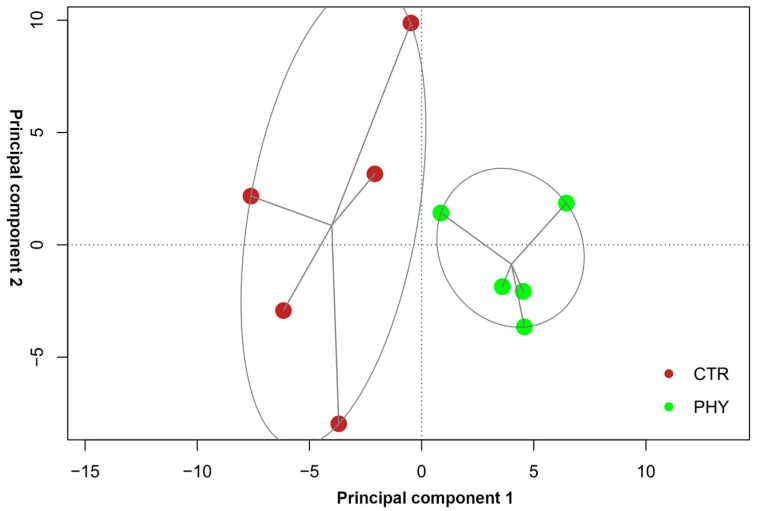
Principal component scatter plot. Distribution of phytogenic treatment (PHY) and control (CTR) samples across the first two principal components. The brown dots represent control, and the green dots point to phytogenic treatment.

**Figure 3 plants-13-01174-f003:**
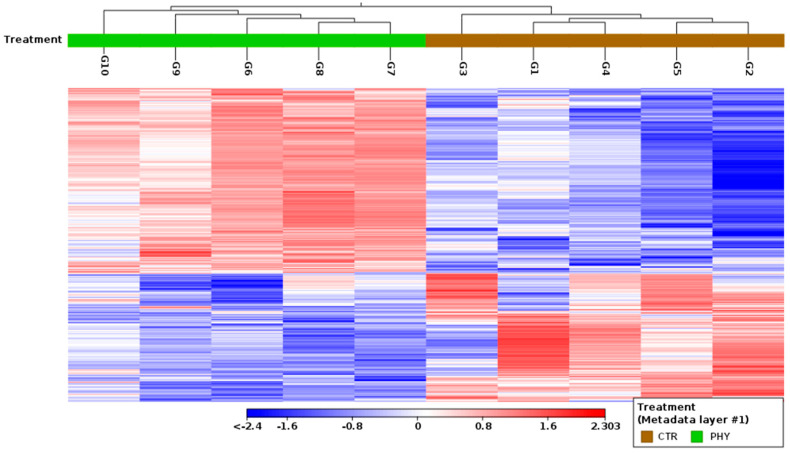
Heatmap of top 100 DE genes (*p* < 0.05, |fold change| > 1.5). Upregulated genes are colored red while downregulated genes are colored blue. Columns differentiate between control and phytogenic treatment.

**Figure 4 plants-13-01174-f004:**
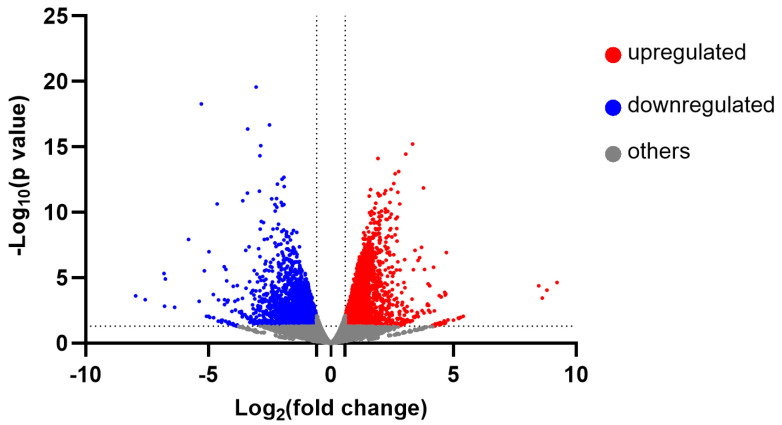
Volcano plot of sequenced genes. All the genes are divided into three categories, including upregulated genes (*p* < 0.05, fold change > 1.5), downregulated genes (*p* < 0.05, fold change < −1.5), and other genes.

**Figure 5 plants-13-01174-f005:**
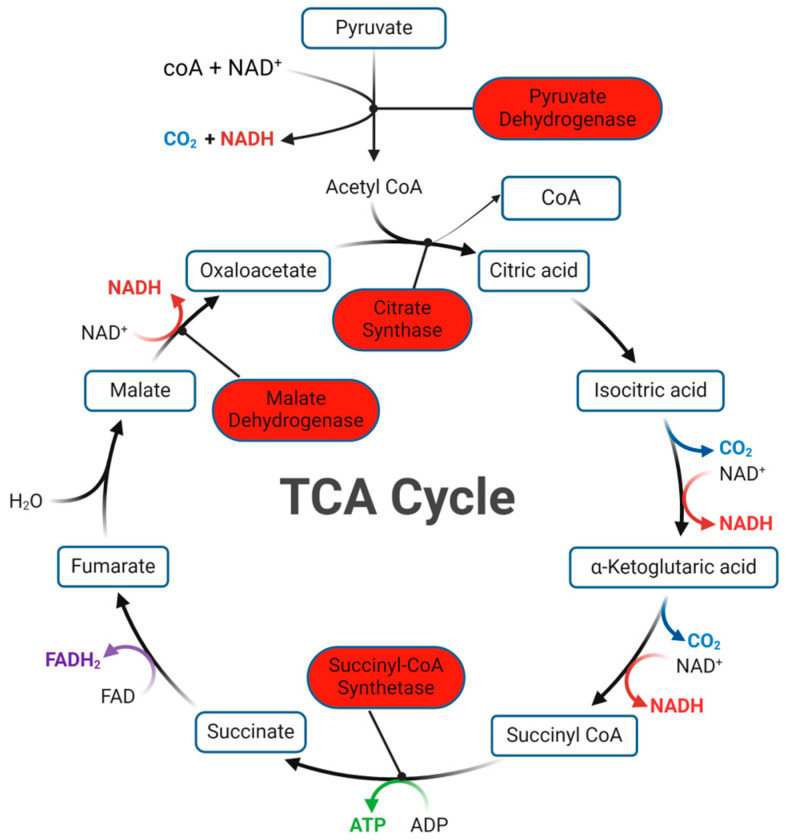
TCA cycle with DE genes after PHY treatment in seedling emergence. Enzymes related to increased genes in PHY are highlighted by red.

**Figure 6 plants-13-01174-f006:**
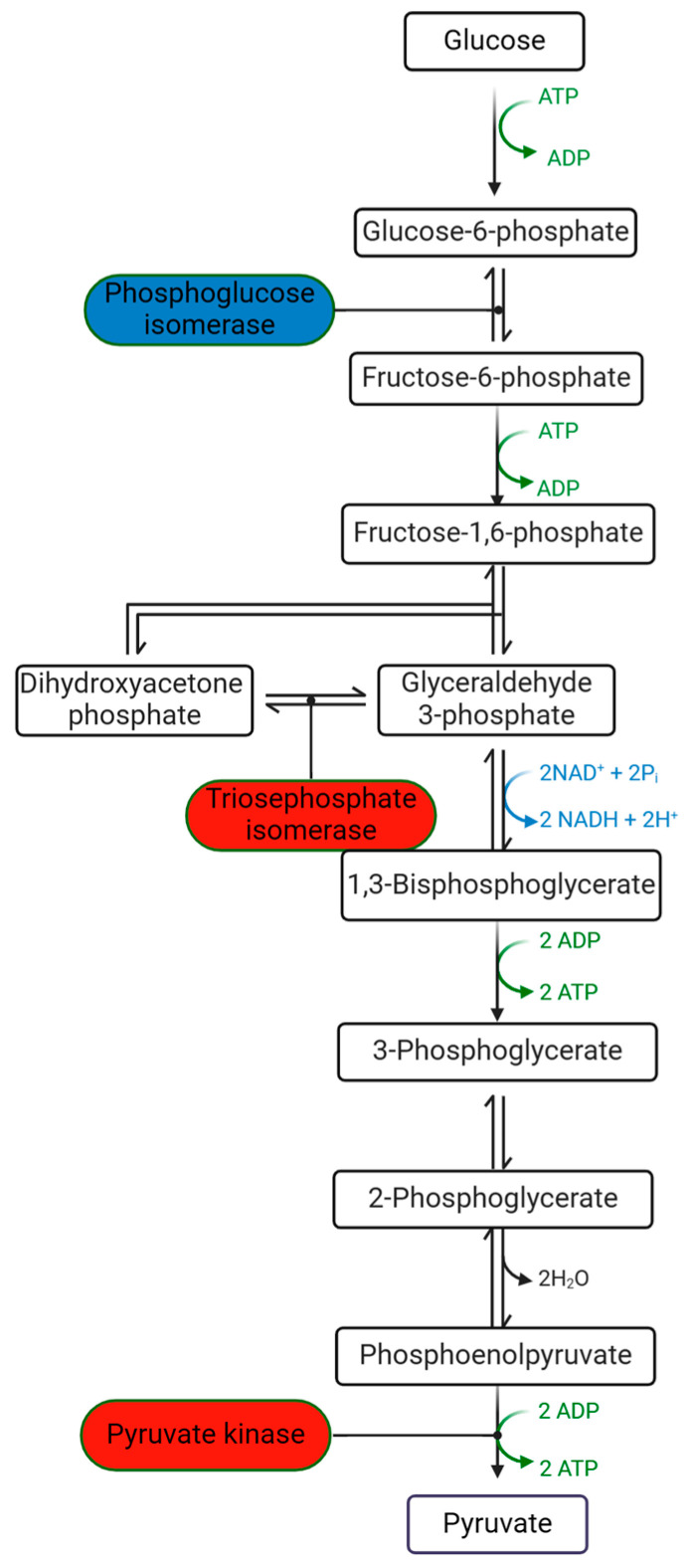
Glycolysis with DE genes after PHY treatment in seedling emergence. The inhibitory and activated enzymes in the phytogen treatment are highlighted by blue and red, respectively.

**Figure 7 plants-13-01174-f007:**
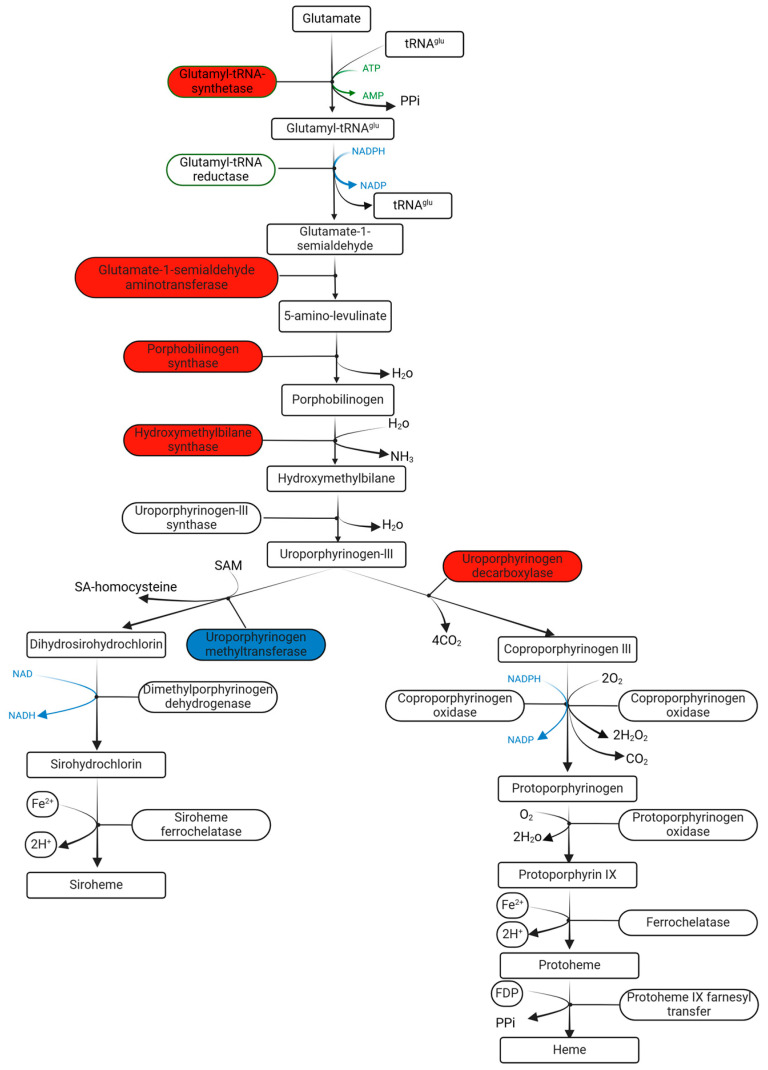
Heme biosynthesis with DE genes after PHY treatment in seedling emergence. Enzymes related to highly expressed genes are highlighted by red and less expressed gene is highlighted by blue.

## Data Availability

Raw sequencing data supporting the results and conclusion made in this study are available in public repositories of the National Center for Biotechnology Information (NCBI) SRA database with accession number PRJNA1068799.

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
