# Peer review of "Transcriptomic Insights: Phytogenic Modulation of Buffel Grass (Cenchrus ciliaris) Seedling Emergence"

_plants, 2024, doi:10.3390/plants13091174_

Round 1
Reviewer 1 Report
Comments and Suggestions for Authors
In this article, Ren et al. have tried to investigate the mechanism of HPY on seed emergence of buffel grass (Cenchrus ciliaris) by means of RNA seq. They studied the influence of different PHY concentrations on seed germination rate and performed RNA-Seq analysis under optimal germination concentration treatment conditions. They focused on DEG analysis of metabolic pathways related to carbohydrate, amino acid and nucleotide synthesis. In the article, the authors also mentioned the effect of endogenous signals, such as hormones, on seed germination. However, in this work, no relevant studies and detailed discussions were carried out, and PHY did not perform a detailed correlation analysis on the expression of genes related to hormone synthesis and transcription, such as ABA and GA. The current experiments could not confirm the main regulatory pathways for PHY affecting seed germination. In addition, this work lacks further real-time fluorescence quantitative PCR validation for genes involved in significant differences.
Below are other comments.
1. Figure 1. Only quantitative data on seedling emergence are given here. Actual photographs showing seedling emergence after treatments at different concentrations would be useful to supplement this information.
2. The effect of PHY on TCA cycle. We know that the TCA cycle is the common pathway capable of breaking down the three major nutrients, sugar, fat and protein, and that glycolysis is also a fundamental metabolic pathway. It is only by increasing the expression of individual genes that metabolic and biosynthetic pathways are affected, and there is no experimental evidence for this yet. Is it possible to determine the amount of carbohydrates, fats and other nutrients i after treatment with optimal concentrations to further demonstrate that PHY treatment has an effect on Cenchrus ciliaris?
3. Line199-207: It is not very accurate to assume that PHY has an upstream and downstream effect on TCA cycling and cellular energy production simply because of the high expression of these four genes, and there is a lack of other experiments to validate this idea.
Reviewer 2 Report
Comments and Suggestions for Authors
In this article the authors analyze the alterations in gene expression in buffel grass under phytogen treatment. This study involves a standard transcriptome analysis, and the experimental design is relatively simple. Here are my major concerns:
1. In the transcriptome analysis section, the author used the rice genome as a reference sequence. In fact, the author can also perform de novo transcriptome analysis, which involves first transferring the transcript and then using the assembled transcript as a reference sequence for gene quantitative analysis. Please explain why the rice genome is used as a reference sequence, whether there is sufficient genetic relationship between rice and buffer grass, and what is the specific alignment rate? If the alignment rate is less than 50%, it cannot be used for gene expression quantification.
2. For transcriptome analysis, it is necessary to select some differentially expressed genes for quantitative PCR validation
3.The subheadings in the results section are too simple, especially 2.2 Quality control. The author should provide sufficient information in the subheadings to facilitate reader reading.
4.The author should provide information of the site where buffel grass seeds were collected, and the variety name, whether it is wild or cultivated.
5. For figures. The buffel grass in upper part of Figure 1 should be deleted. The font size of the X-axis and Y-axis titles in Figure 4 is too large. Figure 5 is missing. The resolution of Figures 6-8 is very low and should be replaced with higher quality images, at the same time, the titles of these figure are too simple and should be rewritten.
Reviewer 3 Report
Comments and Suggestions for Authors
This is a novel and current study in the area of plant production and also plant protection through the application of botanicals where it is necessary to expand the knowledge and methods for the evaluation of the different modes of action of these natural compounds. The article describes in great detail the mechanisms of activation in the plant of germination as well as the metabolic pathways that are activated in the seed by the presence of the botanical compound. The concentration is adequately determined and the method is adjusted to a required statistic. The results of this study will be used for the development of new botanical compounds for growth promotion and metabolic activation, as well as their potential use in the control of pests and diseases through induced resistance in the plant.
Reviewer 4 Report
Comments and Suggestions for Authors
Dear Authors,
below are my comments on the manuscript.
The manuscript explores the impact of a novel phytogenic product containing citric acid, carvacrol, and cinnamaldehyde, on Cenchrus ciliaris seedling emergence.
Title: too long. I ask the authors to shorten it and make it more concise.
Abstract is adequate, but the number of keywords is low. It would be worthwhile for the authors to add 2-3 more keywords
The objectives are not very specific, so I would ask the authors to emphasise better what the aim of the experiment was.
Results.
Fig2: please explain the axis titles within the diagram.
Fig 4: the font size is unnecessary, it should be reduced.
Conclusion: the authors should rewrite the chapter, expand it a bit and add future implications and results.
Round 2
Reviewer 1 Report
Comments and Suggestions for Authors
In the revised manuscript, the authors conducted a preliminary exploration of the mechanism by which PHY affects the seed emergence of buffel grass (Cenchrus ciliaris) and achieved certain research results, there are still certain problems and limitations in data interpretation, gene screening, metabolic pathway analysis, and conclusion promotion, which require further experimental improvement.
1. Line 200-203: The article mentions that carbohydrate metabolism pathways are related to rapid growth processes, but only four genes are significantly overexpressed, without delving into how these genes work together and their specific positions and roles in metabolic pathways.
2. Although the article concludes that PHY has an impact on the pentose phosphate pathway, due to limitations in experimental conditions, further experimental validation is needed to obtain this conclusion.
Reviewer 2 Report
Comments and Suggestions for Authors
I have no other comments.
Reviewer 4 Report
Comments and Suggestions for Authors
Dear Authors, you have corrected the manuscript accordingly, and I accept it for publication.